# One-Step Synthesis of 3D Graphene Aerogel Supported Pt Nanoparticles as Highly Active Electrocatalysts for Methanol Oxidation Reaction

**DOI:** 10.3390/nano14060547

**Published:** 2024-03-20

**Authors:** Xiaoye Wo, Rui Yan, Xiao Yu, Gang Xie, Jinlong Ma, Yanpeng Cao, Aijun Li, Jian Huang, Caixia Huo, Fenghua Li, Yu Wang, Liqiang Luo, Qixian Zhang

**Affiliations:** 1School of Materials Science and Engineering, Shanghai University, Shanghai 200436, China; moqiskull@gmail.com (X.W.); yanrui@shu.edu.cn (R.Y.); mylittlefishes@outlook.com (X.Y.); mjl15385691785@163.com (J.M.); 18652844482@163.com (Y.C.); aijun.li@shu.edu.cn (A.L.); jianhuang@shu.edu.cn (J.H.); huocaixia2021@shu.edu.cn (C.H.); luck@shu.edu.cn (L.L.); 2Shaoxing Institute of Technology, Shanghai University, Shaoxing 312000, China; xiegang@shu.edu.cn (G.X.); fhli@ciac.ac.cn (F.L.); 3State Key Laboratory of Electroanalytical Chemistry, Changchun Institute of Applied Chemistry, Chinese Academy of Sciences, 5625 Renmin Street, Changchun 130022, China; 4School of Horticulture, Jilin Agricultural University, Changchun 130022, China

**Keywords:** graphene aerogel, Pt nanoparticles, 3D architecture structure, one-step hydrothermal self-assembly, electrocatalytic oxidation

## Abstract

Nowadays, two of the biggest obstacles restricting the further development of methanol fuel cells are excessive cost and insufficient catalytic activity of platinum-based catalysts. Herein, platinum nanoparticle supported graphene aerogel (Pt/3DGA) was successfully synthesized by a one-step hydrothermal self-assembly method. The loose three-dimensional structure of the aerogel is stabilized by a simple one-step method, which not only reduces cost compared to the freeze-drying technology, but also optimizes the loading method of nanoparticles. The prepared Pt/3DGA catalyst has a three-dimensional porous structure with a highly cross-linked, large specific surface area, even dispersion of Pt NPs and good electrical conductivity. It is worth noting that its catalytic activity is 438.4 mA/mg with long-term stability, which is consistent with the projected benefits of anodic catalytic systems in methanol fuel cells.. Our study provides an applicable method for synthesizing nano metal particles/graphene-based composites.

## 1. Introduction

Traditional energy utilization methods are difficult to match to the development of modern society, and because of this, how to effectively use energy has become a top priority. At the same time, the traditional utilization methods have also caused a huge amount of water, gas, thermal and noise pollution to our living environment. Fuel cell power generation technology with high energy utilization efficiency without pollution may be just what people have been searching for [1,2,3]. Among all kinds of fuel cells, the direct methanol fuel cell (DMFC) has been documented for its features, such as zero emission [4], high energy density [5], low cost [5], high energy conversion efficiency [6], convenient preparation [7] and other advantages. Although DMFCs have long been regarded as the first choice for green power generation equipment in portable electronic devices [7,8,9,10], its further development is largely hindered by the insufficient activity and relatively high cost of current platinum-based cathode catalysts [11,12,13]. As a noble metal material with excellent catalytic activity and adaptability to a variety of catalytic reactions, the total amount of platinum resources in the earth’s crust is extremely limited [14,15]. Therefore, platinum nanoparticles (Pt-NPs) are usually supported on matrix materials with a high specific surface area [16,17] to achieve maximum activity and minimum consumption of platinum [18,19].

It is believed that in acidic electrolytes, the electrocatalytic oxidation process of methanol on Pt catalysts includes the following steps:CH_3_OH + Pt → PtCH_2_OH + H^+^ + e(1)
PtCH_2_OH + Pt → Pt_2_CHOH + H^+^ + e(2)
Pt_2_CHOH + Pt → Pt_3_COH + H^+^ + e(3)
Pt_3_COH + Pt → PtCO + 2Pt + H^+^ + e(4)
Pt(M) + H_2_O → Pt(M)OH + H^+^ + e(5)
PtCO + Pt(M)OH → Pt + Pt(M) + CO_2_ + H^+^ + e(6)

During the electrooxidation of methanol, methanol adsorbed on the catalyst surface undergoes dissociation and adsorption by the influence of Pt, generating adsorption intermediates and four protons, releasing six electrons. The released electrons were finally transferred to the catalyst surface and discharged. This indicated that steps (1)–(4) occurred at the Pt site and Pt-CO was formed, while the key step in the electrooxidation of methanol was to convert CO from chemical adsorption to CO_2_. The further oxidation of the chemisorbed CO to CO_2_ requires reaction with hydroxyl groups, and step (24) was the process of water activation and chemisorption of hydroxyl groups. Finally, the chemisorbed OH reacted with the chemisorbed CO to form carbon dioxide, making the Pt surface clean, which was proceeded by the next catalytic cycle [20]. From the above mechanism, it can be found that step (1)–(4) may inhibit the whole reaction and affect the usage time and performance, while (5)–(6) was very important for subsequent oxidation. At the same time, the whole process will also have an impact on the efficiency of the battery. In general, the effect of catalyst surface structure on the overall process was not negligible. 

In order to improve the activity of the catalyst and reduce the amount of noble metal used, scientists have tried to develop new carbon materials with good conductivity and high specific surface area, for example, carbon black (CB) [21,22,23], carbon nanotubes (CNT) [24,25,26], mesoporous carbon [27,28,29], etc. Due to the excellent performance of carbon materials, graphene aerogels (GA) born out of graphene materials have also attracted people’s attention. Since the discovery of graphene in 2004, its extremely high specific surface area, single-atom thickness, and excellent physical and chemical properties have made graphene materials the best choice for anode catalysts in DMFCs [30,31,32,33,34,35,36]. However, the stacking of graphene sheets from the effect of van der Waals force hinders them to be used as good supports for fuel cells [37,38]. GA is a porous material with a three-dimensional network structure created by self-assembly and lapping of graphene. This material has an extremely low density of only 0.16 mg/cm^3^, high electrical conductivity and a porous structure used as a strongly adsorbing material [39,40]. So, GA have been used as catalysts [41], adsorbents [42,43,44], electromagnetic shielding materials [45], etc. The applied fields include environmental protection [46], chemical industry, energy storage [47], sensors [48,49] and more.

As well as we know, GA has been prepared by the template method [50], the sol-gel method [51], the hydrothermal assembly method [52], the cross-linking toughening method [53], 3D printing technology [54], etc. It is often necessary to remove the internal residual moisture of the GA material by means of supercritical drying and freeze-drying techniques to obtain a stable GA material at the last step. These technologies need a long experimental period with a low output and high cost [55], which are difficult to achieve and actualize when compared to low cost mass production. Therefore, how to develop a relatively simple, efficient, low cost and reproducible technology for large-scale preparation of GA materials is particularly important. For three-dimensional aerogel materials, reasonable structural design plays an important role in high compressibility and elasticity. There is also the question of how to stabilize the three-dimensional structure of GA without material collapse during the synthesis process. In addition, appropriate catalyst particle size and uniform dispersion are also prerequisites for effective catalytic reactions, and the basic catalytic performance of catalysts is also strongly affected by the synthesis method and the selection of matrix materials [56]. In fact, at present, there are two main ways of loading platinum nanoparticles on the matrix material. One is loading platinum nanoparticles on the finished GA material through physical adsorption [57,58]. The effect of this method is extremely limited, which causes a huge waste of platinum. The solvothermal co-reduction [59] method, a new one-step method, was proposed to prepare the graphene-based nanocatalyst.

In this paper, another new hydrothermal assembly method was introduced through mixing graphene oxide slurry and platinum nanoparticle precursors. General freeze-drying technology was replaced by a freeze–thaw treatment. Three-dimensional graphene aerogel inserted with platinum nanoparticles (Pt/3DGA) was successfully prepared by utilizing the one-step method. In order to evaluate the catalytic performance of Pt/3DGA, it was used as a catalyst for methanol catalytic oxidation.

## 2. Materials and Methods

### 2.1. Materials and Chemicals

Commercial graphite powder (Sinopharm Chemical Reagent Co., Shanghai, China), Potassium chloroplatinate (K_2_PtCl_4_| McLean Co., Shanghai, China), Sodium dodecyl sulfate (SDS| Xilong Chemical Co., Shantou, China), L (+)-ascorbic acid (L-AA| Sinopharm Chemical Reagent Co., Shanghai, China), Carbon black (Carbon Black BP-2000| Cabot Co., Boston, MA, USA), Graphene (Graphene| Xianfeng Nano Material Technology Co., Nanjing, China), Nafion solution (5 wt% isopropanol| DuPont, Wilmington, Delaware), Methanol (CH_3_OH| Sinopharm Chemical Reagent Co., Shanghai, China) and Sulfuric acid (H_2_SO_4_| Sinopharm Holding Chemical Reagent Co., Shanghai, China) were directly used. The water used was purified by the Millipore water purification system (Merck KGaA, Darmstadt, Germany).

### 2.2. Synthesis of Pt/3DGA Catalysts

Graphene oxide (GO) nanosheets were produced by using a modified Hummers method [60]. 43 g GO slurry (11.6 wt%) was mixed with water (57 g) and ultrasonicated for 10 min to obtain a 5 wt% aqueous solution. A mixed solution of K_2_PtCl_4_ (10 mg), 5% SDS aqueous solution (50 mg), AA (20 mg) and 1 g GO aqueous solution (5 wt%) was sonicated for 10 min to form a stable compound solution. Detailed information about the content of Pt/Graphene aerogel, Pt/carbon black and Pt/graphene are stated in Appendix A. Then, at room temperature, the raw materials were stirred by a magnetic stirrer for 20 min until fully foaming, and then reacted at 90 °C for 1 h. The sample was naturally cooled to room temperature, placed in a −4 °C environment for 6 h and then formed the 3D network porous structure of the aerogel. Afterwards, the GA frozen samples were taken out and heated at 85 °C for 6 h. Finally, the samples were rinsed three times with ultrapure water and ethanol, then dried at 80 °C for 2 h to obtain the Pt/3DGA samples. (The experimental parameter settings and experimental details are shown in Appendix A).

### 2.3. Synthesis of Pt/Carbon Black and Pt/Graphene

Pt catalysts supported on carbon black (BP 2000) and graphene were prepared by physical adsorption as control samples. They were labelled as Pt/C and Pt/G, respectively. 40 mg of carbon black and graphene were exposed to 2 mL of ethylene glycol solution containing 10 mg K_2_PtCl_4_. The solutions were further heated at 120 °C for 10 h to obtain the corresponding Pt/C and Pt/G samples.

### 2.4. Electrochemical Characterization

Electrochemical tests were performed on a CHI 660E (CH Instruments, Austin, TX, USA) electrochemical workstation using a standard three-electrode system. A platinum wire was used as the counter electrode. A saturated calomel electrode (SCE) was the reference electrode. The catalyst-coated glassy carbon electrode (GC, 3 mm in diameter) was used as the working electrode. The working electrode was prepared as follows: 4 mg of the catalyst sample was uniformly dispersed in a mixed solution (950 μL of water, 950 μL of ethanol and 100 μL of 5% Nafion) by sonication for 30 min. Subsequently, 5 μL of the above suspension was evenly dropped onto the GC electrode and then it was dried at 40 °C. 

For cyclic voltammetry (CV), a potential voltage ranges from 0.05 V to 1.1 V (vs. SCE) was applied in 0.5 M H_2_SO_4_ containing 1 M methanol solution. Meanwhile, sustained voltammetry cycling was conducted from 0.05 to 1.1 V (vs. SCE) in order to evaluate the long-term stability behavior of the catalyst. The fresh electrolyte solution is used for each electrochemical measurement to ensure reproducible measurement results.

In the control experiments, the electrocatalytic performance of Pt/GA, Pt/C and Pt/G was characterized from the potential range of −0.3 V to 1 V in 0.5 M H_2_SO_4_. In addition, the electrochemically active surface areas (ECSAs) of different catalysts were obtained by analyzing of hydrogen adsorption/desorption peaks.

### 2.5. Characterization Tests

The Brunauere–Emmette–Teller surface area (BET, Micromeritics 3Flex| Micromeritics, Norcross, GA, USA), Scanning electron microscope (SEM, ZEISS Sigma 300| Oberkochen, Germany), X-ray diffractometer (XRD, Bruker D2 Phaser| Bruker, Karlsruhe, Germany), X-ray photoelectron spectroscopy (XPS, Thermo Scientific K-Alpha| Thermo Scientific, Waltham, MA, USA) and Raman microscope (Horiba Lab RAM HR Evolution| Horiba Scientific, Kyoto, Japan) were used, respectively, to characterize the structure and morphology of the product. 

## 3. Results

### 3.1. Synthesis of Materials

The preparation process of Pt/3DGA with 3D structure is shown in Figure 1. First, SDS was added to the homogeneous GO dispersion mixed with K_2_PtCl_4_, and a large number of air bubbles were generated in the dispersion by magnetic stirring. GO sheets are dispersed in the solution due to the electrostatic repulsion from oxygen-containing functional groups on the sheets [61,62]. The GO sheets are independent of each other, and the small molecule K_2_PtCl_4_ can be evenly attached to the defects of GO sheets, thereby forming uniformly dispersed GO-PtCl_4_^2−^ complex in the solution. Next, the homogeneously dispersed GO-PtCl_4_^2−^ composites were hydrothermally treated with reducing agent L-AA to obtain Pt/3DGA. During the hydrothermal reaction, the stable electrostatic balance between the GO sheets is destroyed. Because of the loss of oxygen-containing functional groups, the sheets then prefer to bridge with each other. The air bubbles generated by SDS as a foaming agent can play a supporting role for the GO sheets [63]. Therefore, after the reaction, a stable three-dimensional overall hydrogel gel structure can be formed between the reduced graphene sheets [64]. 

Subsequently, the three-dimensional hydrogel structure of 3DGA is further stabilized through freezing–thawing treatment, which effectively reduces the possibility of the collapse of the three-dimensional structure in reduced graphene sheets during the drying process. Finally, the original three-dimensional hydrogel gel structure is processed by washing and drying to obtain the finished Pt/3DGA. Unlike the common use of freeze-drying technology, our experimental protocol can quickly and efficiently prepare GA samples without the use of complicated instruments.

### 3.2. Structural and Morphological Features

The morphology and structure of the prepared Pt/3DGA were carefully studied. The prepared aerogel materials are light in weight and show a relatively loose porous structure (Figure 2A). SEM images show that Pt/3DGA presents an interconnected three-dimensional porous structure, and the pores range from a few microns to tens of microns (Figure 2B). This porous graphene framework with a rough surface can maximize the exposure of active sites, while the lamellar structure can provide more effective anchor attachment points for catalysts [65]. In fact, the complete single-layer graphene structure has a large surface energy in theory, and is prone to spontaneous aggregation, so it has difficulty existing. Therefore, its surface energy is often reduced through surface wrinkles or edge effects. In the same way, the wrinkles on the surface of 3DGA and the rough, porous graphene skeleton not only provide sufficient sites for large amounts of Pt NPs to be loaded, but also greatly reduce their surface energy, allowing methanol molecules in the solution to be more easily adsorbed on the matrix by Pt NPs to complete a series of catalytic oxidation reactions. As shown in Figure 2C, Pt NPs were uniformly anchored on the surface of 3DGA, indicating a strong interaction between the metal and the aerogel. Nitrogen adsorption/desorption isotherms were used for the determination of textural properties of the support material. 

The BET surface areas (Figure 2D) and pore size (Table 1) of Pt/3DGA were characterized by the N_2_ adsorption/desorption isotherms and the Barrete–Joynere–Halenda (BJH) adsorption model, respectively. It is generally accepted that a high surface area can provide more active sites and mesoporous structures in which 3DGA can improve oxidation of formic acid and methanol. The BET-specific surface area of Pt/3DGA was found as 227.89 m^2^ g^−1^, and its average pore size was found as 13.65 nm. The values of the BET surface area, the average pore size of BJH adsorption and the cumulative volume of pores are given in Table 1. The large porosity and specific surface area of the graphene aerogel network also play an extremely important role in the adsorption of Pt nanoparticles. The SEM results also demonstrated the successful preparation of Pt/3DGA with an interconnected porous network. In addition, with the help of elemental mapping and EDS, it was shown that the Pt/3DGA catalyst is composed of C, O and Pt (Appendix A), and these three elements are uniformly distributed in the sheet.

As shown in Figure 3A, the crystal structure of Pt/3DGA nanocomposites was studied using the XRD method. A sharp diffraction peak around 11° is observed in the image of GO, which is related to the (002) reflection of the carbon hexagonal layered structure of the inner stacked GO. Whereas, the peaks for Pt/3DGA and 3DGA materials moved to a higher angle of ~24°, indicating the complete transformation of original oxygen-containing groups in GO into graphene during the solvothermal reaction. Corresponding to the XRD pattern of Pt/3DGA, the Pt/3DGA nanocomposites also had a C (002) peak, but its intensity was much lower than that of 3DGA, which might be attributed to the introduction of Pt between graphene sheets. In addition, the three characteristic peaks centered at 40.1°, 46.4° and 68.2° were corresponded to (111), (200) and (220) crystal planes of Pt crystals with face-cantered cubic structure (fcc), respectively. As the crystalline Pt nanoparticles loaded on 3DGA had no obvious impurity phase, which was consistent with the previously determined EDS data, this finding might be attributed to the presence of abundant oxygen-containing functional groups of GO, which was beneficial for anchoring the Pt nanoparticles. The XRD results showed that the synthesized Pt/3DGA nanocomposite was composed of 3DGA and pure Pt NPs. The Debye–Scherrer Equation (7) was used to determine the average crystallite size (D) of Pt for the diffraction planes:(7)$$D=\frac{K\lambda }{\beta cos\theta }$$
in which *K* is a dimensionless shape factor and has a value of ~0.9; *λ* is the X-ray wavelength (~1.54178 Å); *β* is the full width at half maximum height (FWHM) expressed in radians and *θ* is the Bragg angle.

On the basis of the full width at half maximum of the diffraction plane, the crystalline size of Pt NPs in nanocomposites was calculated to be 2.37 nm (Table 2).

As a commonly used optical measurement method, Raman spectroscopy is very sensitive to molecular bonding and sample structure, so it is mostly used to identify structural changes and electronic features of carbon-based materials [66]. Figure 3B is the Raman spectra of GO, 3DGA and Pt/3DGA. Two characteristic peaks appeared at around 1350 cm^−1^ and 1600 cm^−1^, which were, respectively, corresponded to the D band and G band of carbon materials. The D band represents the ring breathing vibration mode of the sp^2^ hybridized carbon atom ring in the C atom lattice, showing the defects in the C atom lattice, and its intensity is generally used to measure the disorder of the material structure, while the G band is caused by the stretching vibration between sp^2^ carbon atoms in the C atom crystal, and its intensity is related to the size of the crystal. In addition, it can be observed that there is a peak between 2600 cm^−1^ and 2800 cm^−1^. This peak is the 2D peak, which is the second-order Raman peak of two-phonon resonance and is used to reflect the stacking method of multi-layer graphene.

Here, the D band was corresponded to the defects and disorder induction of 3DGA, while the G band was caused by the ordered sp^2^ carbon atoms [67,68]. What is more, the intensity ratio of the D band and the G band, I(D)/I(G), was used to judge the integrity of the graphene material structure. The I(D)/I(G) of the obtained 3DGA was 1.07 without adding platinum precursor, which was higher than that of GO (0.93), indicating the disordered graphene layer of 3DGA. The I(D)/I(G) of Pt/3DGA (1.25) was much higher than that of 3DGA (1.07) and most previous studies [57,69,70]. 

Then, we compared the SEM images of GO, 3DGA and Pt/3DGA (Appendix A). It can be clearly seen that with the increase of the degree of reduction and the loading of Pt NPs, the surface roughness and non-linearity of the material increases. The degree of ordering is significantly improved, thus gradually showing its high porosity, low density and high specific surface area. This is consistent with the fact that the peak ratio I(D)/I(G) becomes higher, meaning there is a higher of degree of disorder of the material. This might be related to the intercalation of Pt nanoparticles into 3DGA defects, bonding with single-layer graphene nanosheets [58]. The loading of platinum nanoparticles further induced more defects to 3DGA, which were also contributed to the disordering of the 3DGA samples. In addition, the defects in 3DGA could also provide active sites for the electron transfer between Pt nanoparticles and graphene [71]. Given this, it would be a good catalyst for methanol electrocatalytic oxidation.

XPS was further used to characterize the elemental composition of the Pt/3DGA catalyst. Figure 3C shows the high resolution C 1 s spectrum of Pt/3DGA. After deconvolution, four different peak components are obtained, namely C-C (284.7 eV), C-O (285.6 eV), C=O (287.5 eV) and O-C=O (290.1 eV). Compared to the C 1 s spectrum of GO, the peak intensity of oxygen-containing groups in Pt/3DGA is further reduced, revealing that GO has been successfully converted into graphene sheets during the hydrothermal reduction process. Furthermore, the Pt 4f spectrum in Figure 3D reveals two pairs of doublets (72.3 and 76.0 eV) of Pt inside Pt/3DGA, which is higher than that of the metallic atomic state Pt (71.4 and 74.7 eV). This indicates that a small number of PtO states are present in the graphene sheets and confirms the successful loading of Pt nanoparticles on the graphene sheets in the aerogel.

### 3.3. Electrochemical Properties

The electrochemical performance of Pt/3DGA in a 1 M H^+^ environment was evaluated by CV to estimate the ECSA of the Pt catalyst. Figure 4A is a typical CV image of Pt/3DGA, Pt/C and Pt/G samples. In Figure 4A, we can clearly distinguish the hydrogen region (−0.2 V~0.1 V), double electric layer region (0.1 V~0.3 V) and oxygen region (0.3 V~1.1 V). There is a pair of hydrogen adsorption/desorption peaks in the hydrogen region, and the area of the hydrogen adsorption/desorption peaks is often used to evaluate the active surface area of electrocatalysts. 

In addition, comparing the reduction peak potential of the catalyst in the oxygen region can also be used to measure the quality of the catalytic performance. The more the reduction peak potential shifts toward the cathode, the better the electrocatalytic activity on methanol. From −0.2 V to 0.1 V (vs. SCE), there were two obvious hydrogen adsorption/desorption peaks on the CV curves, which could be used to calculate the ECSA of different catalysts. H^+^ ions were reduced during the forward scan. An anodic current was generated by desorption of the adsorbed hydrogen during the corresponding reverse scan. Through calculation, the ECSA values of Pt/3DGA, Pt/C and Pt/G were 41.4 m^2^/g, 13.9 m^2^/g and 21.4 m^2^/g, respectively (Appendix A). So, the ECSA value of the Pt/3DGA catalyst was about 2.98 and 1.93 times that of the Pt/C and Pt/G. This confirms that hydrogen adsorption and desorption is easier on Pt/3DGA composite modified electrodes. Pt is better dispersed on carbon carriers, such as GA. Under the same loading capacity, the active specific surface area is larger. In addition, the charge and discharge current of Pt/3DGA in the electric double layer region is larger, which also shows that the GA-based material changes the adsorption and desorption properties of hydrogen on the Pt surface and generates more oxygen-containing active species, causing the oxide reduction peak potential to move toward the cathode (Pt/3DGA: 0.50 V; Pt/G: 0.53 V; Pt/G: 0.51 V).

Therefore, the Pt/3DGA could provide more active sites for methanol oxidation and might have better electrocatalytic activity.

Subsequently, methanol catalytic oxidation tests were conducted in 1 M H^+^ and 1 M methanol solutions at a scanning rate of 50 mV s^−1^ under a temperature of 25 °C. Figure 4B shows the CV curves of different catalysts in methanol electrooxidation. In the Figure, the CV curves of all catalysts have two obvious strong methanol oxidation peaks, while the hydrogen adsorption peak is suppressed. This is caused by the competitive adsorption between methanol and hydrogen on the catalyst surface. Methanol is preferentially adsorbed and then oxidized, while hydrogen adsorption is suppressed. Among them, the peak corresponding to the potential of about 0.47 V is the forward sweeping peak, which is the oxidation peak of methanol, while the peak corresponding to around 0.72 V is the reverse sweeping peak, which is the oxidation peak of the methanol oxidation intermediate product in the electrocatalytic process, such as CO, etc. The peak current density of the oxidation peak and its corresponding peak potential can be used to measure the activity of the catalyst in the catalytic oxidation of methanol, which means that greater peak current density and smaller corresponding peak potential equals a stronger electrocatalytic performance. Then, the figure shows that the Pt/3DGA catalyst has a lower methanol oxidation onset potential, indicating that the intermediates after oxidative removal of carbon groups are prone to occur on Pt/3DGA catalysts. At the same time, compared to other catalysts (Pt/C: 153.1 mA/mg; Pt/G: 244.0 mA/mg), Pt/3DGA has the highest forward anode peak current density (438.4 mA/mg). The peak current densities of the three catalysts from large to small are Pt/3DGA > Pt/G > Pt/C, and the peak potentials from large to small are Pt/C (0.741 V) > Pt/3DGA (0.703 V) > Pt/G (0.698 V). Although the peak potential of Pt/3DGA is in the middle, its peak current density (438.4 mA/mg) is significantly higher than that of Pt/G and Pt/C catalysts. These experimental results show that the reaction kinetics of the electrocatalytic oxidation of methanol on the Pt/3DGA electrocatalyst are significantly improved, and the starting potential is negatively shifted by about 150 mV compared to the other two catalysts. It can be judged that the activity improvement is due to the synergistic effect between Pt and 3DGA carrier. 3DGA is suitable as a carrier for electrocatalysts.

The methanol catalytic oxidation performance of several recently studied Pt-based carbon material catalysts is shown in Table 3.

Compared to various carbon-based Pt catalysts prepared in recent studies [72,73,74,75,76,77], the peak current density of the methanol electrocatalyst in this study is large (438 mA/mg). The electrocatalytic oxidation reaction has resulted in notable improvements in kinetics and an obvious negative shift in the oxidation’s starting potential. The catalysts on 3DGA support have a good performance for Pt-based electrocatalysts towards methanol oxidation.

The long-term stability of Pt/G and Pt/3DGA catalysts was investigated for methanol electrooxidation by using continuous CV cycles (Figure 5). After a thousand cycles, the forward peak current density of Pt/3DGA dropped by around 20% from its initial value, whereas Pt/G’s forward peak current density decreased by approximately 29.8%. It was particularly noteworthy that the mass activity of Pt/3DGA was higher than that of Pt/G (Figure 5C), implying that the stability of the Pt/3DGA catalyst was significantly higher too.

## 4. Discussion

In summary, the Pt/3DGA was successfully prepared by using a simple and fast one-step hydrothermal self-assembly method. Using the simple method, we not only reduce the cost compared to using freeze-drying technology, but also optimize the loading method of nanoparticles. Meanwhile, the Pt/3DGA catalyst has a unique loose and interconnected porous network structure. It has a large specific surface area (227.89 m^2^ g^−1^) and high porosity. The average pores of the resulting material can reach 13.65 nm. At the same time, a uniform loading of Pt nanoparticles was achieved with a diameter of approximately 2.37 nm, which also makes Pt/3DGA have excellent methanol electrocatalytic activity. The peak current density of Pt/3DGA is 438.4 mA/mg, which is higher than that of most reports. The electrocatalytic activity and long-term stability of the Pt/3DGA catalyst is much better than that of traditional catalysts. After one thousand cycles, the forward peak current density of Pt/3DGA dropped by roughly 20% from its starting value. 

Considering the good performance of the Pt/3DGA catalyst, it has great application potential in fields such as fuel cells, supercapacitors, photocatalysis and sensors.

## Figures and Tables

**Figure 1 nanomaterials-14-00547-f001:**
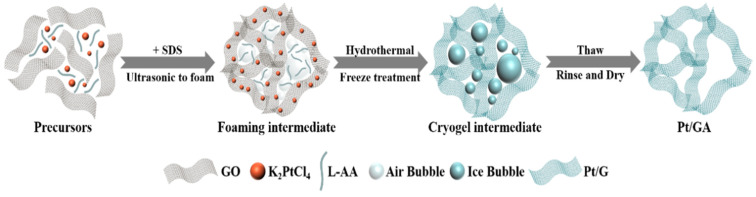
Schematic mechanism for the preparation of Pt/3DGA catalyst.

**Figure 2 nanomaterials-14-00547-f002:**
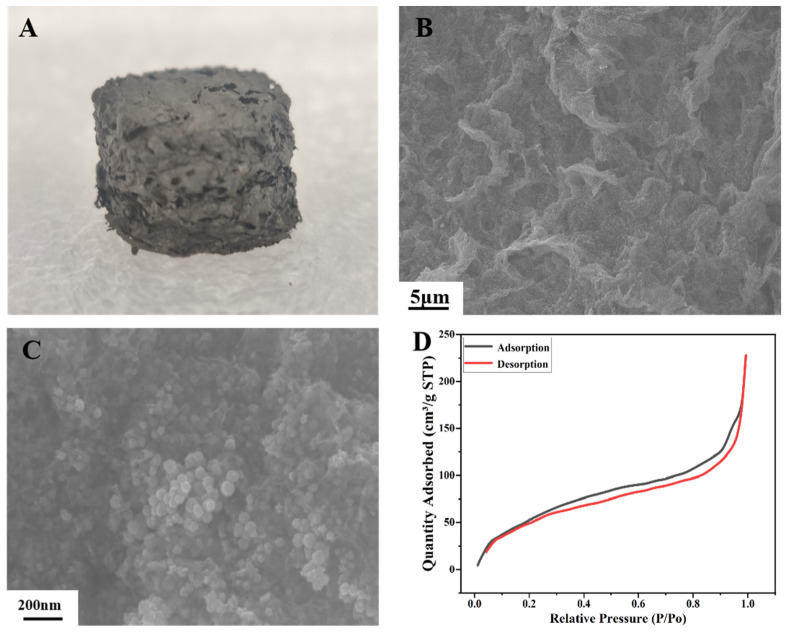
(**A**) A photo of the Pt/3DGA product; (**B**,**C**) SEM images of the Pt/3DGA at different magnifications; (**D**) nitrogen adsorption/desorption isotherms of the Pt/3DGA.

**Figure 3 nanomaterials-14-00547-f003:**
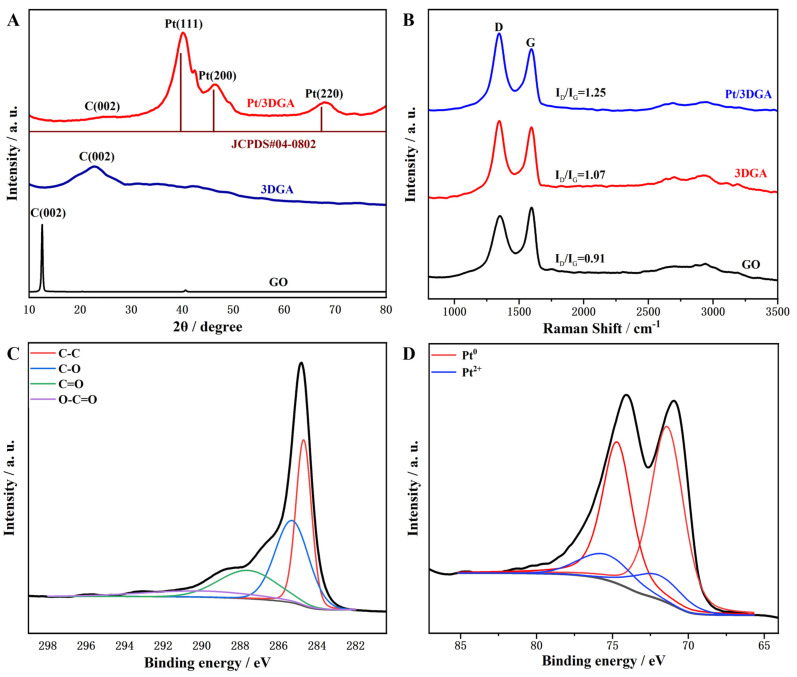
**(A**) XRD spectra of GO, 3DGA and Pt/3DGA; (**B**) Raman spectra of GO, 3DGA and Pt/3DGA; (**C**) C 1s and (**D**) Pt 4f narrow XPS scan of Pt/3DGA (The original Pt/3DGA XPS result is shown as the black line).

**Figure 4 nanomaterials-14-00547-f004:**
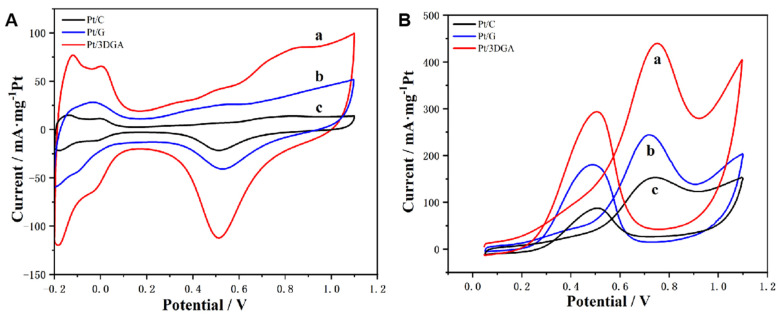
(**A**) CVs of the Pt/3DGA (a), Pt/G (b) and Pt/C (c) composites in a 0.5 M H_2_SO_4_ solution. (**B**) CVs for methanol oxidation reaction catalyzed by Pt/3DGA (a), Pt/G (b) and Pt/C (c) composites in the mixture solution containing 0.5 M H_2_SO_4_ and 1 M CH_3_OH at a scanning rate of 50 mV s^−1^ under a temperature of 25 °C.

**Figure 5 nanomaterials-14-00547-f005:**
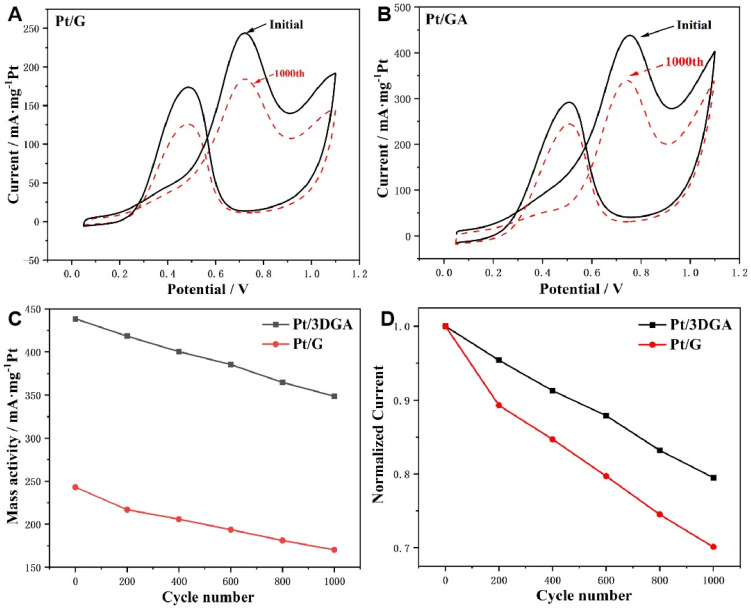
CV of Pt/G (**A**) and Pt/3DGA catalyst (**B**) in a solution of 0.5 mol L^−1^ H_2_SO_4_ containing 1 mol L^−1^ CH_3_OH at a scanning rate of 50 mV s^−1^ under a temperature of 25 °C. Mass activity of Pt/G and Pt/3DGA catalyst with cycle numbers during stability test (**C**). Relationship of normalized peak current density and cycle numbers for Pt/G and Pt/3DGA catalysts (**D**).

**Table 1 nanomaterials-14-00547-t001:** Pore structure parameters and surface areas of Pt/3DGA.

Material	BET Surface Area (m^2^ g^−1^)	Pore Size (nm)	Pore Volume (cm^3^ g^−1^)
Pt/3DGA	227.89	13.65	0.279

**Table 2 nanomaterials-14-00547-t002:** XRD data and crystallite size of Pt NPs.

Sample	Diffraction Plane	FWHM (Radian)	Peak Position (Degree)	Crystallite Size: D (nm)
Pt	(111)	0.05618	20.04	2.63
	(200)	0.05083	23.50	2.97
	(220)	0.11001	33.95	1.52

**Table 3 nanomaterials-14-00547-t003:** The methanol catalytic oxidation performance of our material and other reported materials.

Material	ECSA, m^2^/g(Pt)	E_onset_ CH_3_OH, V ^1,2^	E_peak_ CH_3_OH, V ^1,2^	I_peak_ CH_3_OH, A/g(Pt) ^1,2^	Ref.
Pt/C	76.3	0.386	0.656	~350	[72]
Pt/2rGO-1CNT	468.04	0.360	0.573	359.18	[73]
Pt/Graphene–MWCNTs	88.76	0.253	0.653	168.25	[74]
Pt/C/graphene	70.4	0.176	0.616	405.3	[75]
Pt/N-MWCNTs	94.6	0.196	0.626	539.4	[76]
Pt/e-RGOSWCNT (0.6)	13.31	0.417	0.75	191.7	[77]
Pt/3DGA	41	0.379	0.703	438.4	This study

^1^ Potentials vs. SCE. ^2^ Solution: 0.5 mol L^−1^ H_2_SO_4_ + 1 mol L^−1^ CH_3_OH at scanning rate of 50 mV s^−1^.

## Data Availability

All data included in this study are available upon request by contact with the corresponding author.

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
