# Peer review of "One-Step Synthesis of 3D Graphene Aerogel Supported Pt Nanoparticles as Highly Active Electrocatalysts for Methanol Oxidation Reaction"

_nanomaterials, 2024, doi:10.3390/nano14060547_

Round 1

Reviewer 1 Report

Comments and Suggestions for Authors

In this manuscript, the authors propose platinum nanoparticles-supported graphene aerogel (Pt/3DGA) as an electro catalyst for the methanol oxidation reaction. The endeavour to create innovative and cost-effective materials for methanol oxidation is praiseworthy. However, several aspects require meticulous consideration before deeming the manuscript suitable for publication. The following comments are intended to guide the authors in refining their work:

1.      The mechanism of your work is crucial for understanding. It's recommended to provide a graphical representation of the mechanism involved in your study. This will aid readers in comprehending the processes and reactions described in the manuscript.

2.      Providing more context or detail about why specific temperatures and durations were chosen for the reaction and cooling steps would enhance the reproducibility and understanding of the procedure. Explaining the rationale behind these parameters will assist readers in replicating the experiment effectively.

3.      Authors have stated that the porous graphene framework with a rough surface can maximize the exposure of active sites. It would be beneficial to elaborate on the role of the rough surface in facilitating this process.

4.      Incorporate information about the pore size of Pt/3DGA within Figure-2C. Additionally, it's suggested to explain the exact pore size in the text instead of just mentioning a few. Providing detailed information about the pore size will contribute to a comprehensive understanding of the material's characteristics.

5.      Author described about 3A as Raman image and 3B as XRD image. Author suggested to rectify 186th, 196th, and 205th line for accuracy and clarity. Ensure that the descriptions match the content accurately to avoid confusion among readers.

6.      In Figure 3B, there is a small shift in the plot between 2500 and 3000 cm-1, that should be justified. Try to explain about 2D band and how it affect in the reaction kinetics. Also describe the nature of D-band and G-band.

7.      Correlate Raman shifts with SEM images to explain the structural changes due to disorder. This correlation will provide insights into how disorder affects the material's properties and performance.

8.      Revise Figure 3 to ensure proper alignment and uniform font size. Ensure that all figures are clearly presented and easily distinguishable, avoiding any merging of images or text.

9.      Author suggested to provide a JCPDS Card in Figure 3A to aid in understanding the peaks in XRD. Additionally, maintain a uniform font size for all markings to improve clarity and readability.

10.   Provide a description of the oxidation and reduction peak shifts in figure 4A and 4B. Explaining the significance of these peaks will enhance the interpretation of the electrochemical data presented in the figures.

11.   Author suggested to conduct a thorough comparison of the work with recently published articles and present the findings in a scientific table. Ensure that all tables adhere to scientific formatting standards to facilitate easy comparison and interpretation of the results.

12.   Author suggested to Check Spelling and Grammar, addressing minor spelling and grammatical errors is imperative to enhance the overall readability and professionalism of the manuscript.

13.   In light of the authors' suggestion to emphasize significant advances and innovations, the Conclusion section becomes pivotal. The optimal Conclusion should include:

• A highlight of your hypothesis, new concepts, and innovations.
• A summary of key improvements compared to findings in the literature [provide a couple of references to indicate key improvements].
• Your vision for future work.

Comments on the Quality of English Language

NA

Author Response

Dear reviewers,

Thanks  for your suggestions. Please see the responses in the attachment.

regards,

Qixian Zhang

Reviewer 2 Report

Comments and Suggestions for Authors

The presented article “One-step Synthesis of 3D Graphene Aerogel Supported Pt Nanoparticles as Highly Active Electrocatalysts for Methanol Oxidation Reaction” is devoted to a new approach to the synthesis of Pt/C catalysts on a special graphene support with a 3D structure. This article has a novelty and original approach to the synthesis of catalysts important for hydrogen energy. However, a number of points should be noted:

1) It is necessary to provide information on the area and porosity of the resulting carrier, according to low-temperature gas sorption (BET method)

2) Why is the average size of platinum crystallites not calculated using the Scherrer formula based on XRD data? This information must be provided.

3) The activity of the resulting catalyst must be compared with a standard commercial Pt/C analogue.

4) The authors obtained a material with large platinum particles (about 50 nm), while commercial analog materials and materials described in the literature typically have sizes of Pt in the range of 2–3 nm, with a significantly larger surface area.

5) The part concerning the activity of materials in the methanol oxidation reaction is not described in sufficient detail, how activity was determined, how it differs numerically for different materials, and the reasons for these differences. In addition, it is important to carry out steady-state measurements at different potentials to understand how the current changes over time or measure pseudo-steady-state curves with a slow sweep speed for a more correct assessment of activity. In addition, current output potentials or current values at different potentials are often compared.

6) The CO-stripping method must be additionally applied to the studied catalysts, since this will additionally make it possible to independently verify the ESA value and obtain information about the state of the surface and the features of CO oxidation on the surface, which is important for the oxidation of methanol.

Author Response

Dear reviewers,

Thanks for your suggestions. We have addressed your questions in the attached file.

regards,

Qixian Zhang

Round 2

Reviewer 2 Report

Comments and Suggestions for Authors

The authors answered questions, added additional information, and corrected most comments.

But regarding question 4

«The authors obtained a material with large platinum particles (about 50 nm), while commercial analogues and materials described in the literature typically have Pt sizes in the range of 2–3 nm, with a significantly larger surface area”

The authors' comments are constructed. The fact is that the largest company ESA determines the size of metal nanoparticles according to SEM or TEM data, the average crystallite size according to XRD data is usually lower (in this case, in this work), since an individual nanoparticle can consist of several crystallites, and this the parameter correlates worse. with the largest ESA.

Author Response

(The authors gave the same response as above.)
